# Electron-Beam-Killed *Staphylococcus* Vaccine Reduced Lameness in Broiler Chickens

**DOI:** 10.3390/vaccines12111203

**Published:** 2024-10-23

**Authors:** Anna L. F. V. Assumpcao, Komala Arsi, Andi Asnayanti, Khawla S. Alharbi, Anh D. T. Do, Quentin D. Read, Ruvindu Perera, Abdulkarim Shwani, Amer Hasan, Suresh D. Pillai, Robin C. Anderson, Annie M. Donoghue, Douglas D. Rhoads, Palmy R. R. Jesudhasan, Adnan A. K. Alrubaye

**Affiliations:** 1Department of Poultry Science, University of Arkansas, Fayetteville, AR 72701, USA; af072@uark.edu (A.L.F.V.A.); aasnayan@uark.edu (A.A.); ka030@uark.edu (K.S.A.); ad086@uark.edu (A.D.T.D.); rperera@uark.edu (R.P.); 2Poultry Production and Product Safety Research Unit, United States Department of Agriculture-Agricultural Research Service (USDA-ARS), Fayetteville, AR 72701, USA; komala.arsi@usda.gov (K.A.); annie.donoghue@usda.gov (A.M.D.); 3Cell and Molecular Biology Program, University of Arkansas, Fayetteville, AR 72703, USA; ahasan@uark.edu (A.H.); drhoads@uark.edu (D.D.R.); 4United States Department of Agriculture-Agricultural Research Service (USDA-ARS), Southeast Area, Raleigh, NC 27606, USA; quentin.read@usda.gov; 5Southeast Poultry Research Laboratory, U.S. National Poultry Research Center, United States Department of Agriculture-Agricultural Research Service (USDA-ARS), Athens, GA 30605, USA; abdulkarim.shwani@usda.gov; 6Department of Veterinary Public Health, College of Veterinary Medicine, University of Baghdad, Baghdad P.O. Box 1417, Iraq; 7National Center for Electron Beam Research, Texas A&M University, College Station, TX 77843, USA; suresh.pillai@ag.tamu.edu; 8Food and Feed Safety Research, United States Department of Agriculture-Agricultural Research Service (USDA-ARS), College Station, TX 77843, USA; robin.anderson@usda.gov

**Keywords:** vaccination, lame, *Staphylococcus aureus*, *Staphylococcus agnetis*, lameness prevention

## Abstract

Broiler chicken lameness caused by bacterial chondronecrosis with osteomyelitis (BCO) is presently amongst the most important economic and animal welfare issues faced by the poultry industry, and the estimated economic loss is around USD 150 million. BCO lameness is associated with multiple opportunistic bacterial pathogens inhabiting the respiratory and gastrointestinal tracts. In cases of immune deficiency resulting from stress, injury, or inflammation of the tissue, opportunistic pathogens, mainly *Staphylococcus* spp., can infiltrate the respiratory or gastrointestinal mucosa and migrate through the bloodstream to eventually colonize the growth plates of long bones, causing necrosis that leads to lameness. This is the first report of developing a *Staphylococcus* vaccine against BCO lameness disease in broiler chickens. Electron beam (eBeam) technology causes irreparable DNA damage, preventing bacterial multiplication, while keeping the epitopes of the cell membrane intact, helping the immune system generate a more effective response. Our results show a 50% reduction of lameness incidence in the eBeam-vaccinated chicken group compared to the control. Additionally, the eBeam-vaccinated chickens present higher titer of anti-*Staphylococcus* IgA, signifying the development of an efficient and more specific humoral immune response. Our data establish the eBeam-killed *Staphylococcus* vaccine as an effective approach to reducing the incidence of lameness in broiler chickens.

## 1. Introduction

Bacterial chondronecrosis with osteomyelitis (BCO) is considered a common cause of lameness in broiler chickens, posing significant economic and animal welfare issues in the commercial broiler industry by causing huge revenue losses annually [1,2,3]. The incidence of lameness typically ranges from 1 to 5%; however, during episodic outbreaks, lameness cases can reach 15% [1,2,4,5,6]. BCO lameness impacts poultry revenue by decreasing weight gain and increasing feed conversion ratios, resulting in higher production costs for farmers and increasing carcass condemnation rates in processing plants [7,8].

BCO lesions arise from the formation of microfractures and clefts associated with rapid increase in body weight, leading to focal ischemia, creating a favorable environment for bacterial colonization [1,3,6]. Disruptions in the gastrointestinal and respiratory tracts, such as damaged or immature mucosa, facilitate bacterial translocation. This process allows bacteria to infiltrate through loose or damaged epithelial tight junctions and enter the bloodstream. Bacteria that can survive in the blood can then colonize the proximal growth plates of the long bones [9,10,11]. BCO lesions are more common in the proximal end of the femur and tibiotarsus than in the proximal metatarsus, distal femur and tibiotarsus, proximal humerus, and vertebrae [4,6,12]. Affected birds present a characteristic limping gait that typically is accompanied by the use one or both wing tips for support during locomotion, hip flexion, and/or loud vocalization when using the affected limb [4].

*Staphylococcus* spp., *Enterococcus* spp., and *Escherichia coli* are the most predominant cultures isolated from BCO lesions [3,10,13]. The members of the genus *Staphylococcus*, including *S. aureus*, *S. agnetis*, *S. cohnii*, *S. saprophyticus*, *S. hyicus*, *S. simulans*, and *S. lentus*, were the principal causative agents of BCO lameness identified in our research studies [10,13]. Additionally, these bacteria are a leading cause of infections in children, elderly, and prosthetic device users, causing complications due to the development of multiantibiotic resistance [14,15,16,17]. A compelling method to mitigate these pathogens is the development of vaccines. Several studies have tested vaccines against *Staphylococcus* using target-specific proteins, but these vaccines are inefficient in stimulating an adequate immune response [16,17,18,19]. In contrast, a live-cell *S. aureus* vaccine successfully generated an efficient immune response against *Staphylococcus* infection in a murine model when administered intravenously, showing the importance of using whole-cell vaccines to create an efficacious and long-lasting immune response [20].

Numerous studies have explored the potential of including feed additives such as probiotics, prebiotics, organic trace minerals, and other feed supplements to alleviate the incidence of BCO lameness in broiler chickens [6,21,22,23,24,25,26]. However, these intervention measures may only be able to partially delay or attenuate the development of BCO lameness in broiler chickens. This necessitates the development of novel alternatives to prevent the occurrence of BCO in broiler chickens. An interesting alternative to decrease the incidence of BCO lameness in broiler chickens would be the use of a killed-whole-cell bacterial vaccine, but, at the moment, commercial vaccines are not available [17,23].

Standard methods available to develop inactivated whole-cell vaccines are incubation with formaldehyde or β-propiolactone. However, these techniques are time-consuming and present high variability, requiring extensive quality control procedures. The chemicals used in these vaccine preparations are highly toxic, necessitating elevated biosafety levels for production [27]. Additionally, these chemicals can negatively impact vaccine efficiency by modifying the pathogen’s surface antigens due to cross-linking of protein structures, potentially hindering development of a competent immune response [28,29]. Alternatively, ionizing radiation, such as an electron beam (eBeam), causes multiple double-strand DNA breaks, preventing DNA replication and bacterial multiplication without damaging the cell surface membrane proteins [30]. eBeam technology has been employed worldwide for food pasteurization, decontamination [31], and vaccines [32,33,34]. Therefore, this study aimed to develop killed-whole-cell vaccines using eBeam technology to protect broiler chickens from *Staphylococcus* infections, which is one of the main causes of clinical lameness in broiler chickens. This is the first report of developing a *Staphylococcus* vaccine against BCO lameness disease in broiler chickens.

## 2. Materials and Methods

### 2.1. Bacteria and Culture Condition

We used field-isolated strains of *S. aureus* [13] and *S. agnetis* [10] from lame broiler chickens. The two strains were inoculated separately in tryptic soy broth (TSB, BD Diagnostics, Sparks, MD, USA) and incubated overnight at 37 °C. The overnight grown cultures were mixed in equal portions and centrifuged to remove the spent media. The bacterial pellet was resuspended in TSB at a concentration of ~1 × 10^8^ CFU/mL for vaccine preparation.

### 2.2. Formalin-Killed Vaccine Preparation

Formalin-killed vaccine was prepared as described previously by Kaminski et al., 2014 [35]. Briefly, overnight grown culture was resuspended in TSB at a concentration of ~1 × 10^8^ CFU/mL, and formalin solution (Electron Microscopy Science, Hatfield, PA, USA) was added with a final concentration of 0.6% (*v*/*v*) and incubated with agitation of 200 RPM at room temperature for 48 h. The solution was centrifuged at 6000× *g* for 30 min at 4 °C. The pellet was resuspended in phosphate buffered saline (PBS) and centrifuged at 6000× *g* for 30 min at 4 °C. Finally, the bacterial cell pellet was resuspended in TSB and stored at 4 °C until further use.

### 2.3. eBeam-Killed Vaccine Preparation

The overnight culture was resuspended in TSB at a concentration of ~1 × 10^8^ CFU/mL and was shipped in refrigerated conditions to the National Center for Electron Beam Research at Texas A&M University, College Station, TX, USA. The samples were packed and shipped according to USDA-APHIS, Department of Transportation, and IATA regulations. The eBeam irradiation was performed using a high-energy (10 MeV), 15 kW (power) linear accelerator at a dose rate of approximately 3000 Gy/s. The *Staphylococcus* cultures were exposed to a target dose of 10 kGy (kiloGrays). The delivered dose was measured using alanine dosimeters analyzed with a Bruker E-scan spectrometer (Bruker, Billerica, MA, USA). Aliquots of the irradiated bacterial cells were utilized for tests to confirm inactivation, verification of membrane integrity, and vaccination purposes.

### 2.4. Characterization of eBeam-Killed Staphylococcus Cells

To verify the inability of eBeam-killed bacterial cells to replicate, aliquots of irradiated cells were serially diluted in TSB and plated on CHROMagar Staph aureus (CHROMagar™, Paris, France) agar in parallel. Tubes and plates were incubated at 37 °C overnight and at room temperature for 7 days. The eBeam-killed and untreated bacterial cells were also stained with a LIVE/DEAD BacLight Bacterial Viability Kit (Invitrogen™, Waltham, MA, USA) to determine the integrity of the cell membranes. Briefly, 100 µL of bacterial cells was added to 900 µL of PBS. Then, 5 µL of stain (SYTO 9 + propidium iodide) was added to solution and incubated for 15 min at room temperature protected from light. Then, 100 µL of stained bacterial cell solution was transferred to a 12-well glass bottom plate and examined using fluorescence microscopy. Viable cells would fluoresce green, and non-viable cells (with compromised cell membrane) would fluoresce reddish orange.

### 2.5. In Ovo Vaccination

Three hundred embryonated eggs from a local hatchery were incubated at 37 °C and 85–87% humidity. On day 18, eggs were candled for viability before *in ovo* injection with either a vehicle sham, a formalin-killed vaccine, or an eBeam-killed vaccine (~10^6^ CFU/embryo). Injected eggs were transferred to the hatcher cabinet. This vaccine administration simulated *in ovo* vaccination practices of commercial hatcheries [36]. The large end of the eggs was wiped with 70% ethanol and gently scored with an 18-gauge needle. The vaccine or sham (100 µL) was administered into the amniotic cavity using a 1 mL tuberculin syringe and a 25-gauge needle equipped with a modified needle guard to limit all injections to a depth of 3 cm. Following injection, each egg was sealed with melted paraffin using a cotton swab.

### 2.6. Lameness Trial

Animal procedures were approved by the University of Arkansas Institutional Animal Care and Use Committee (Protocol #21148). Two hundred and ten one-day-old chicks were placed into floor pens containing fresh pine shavings divided into three groups: control, formalin-killed vaccine, and eBeam-killed vaccine groups (two replicate pens per group). Birds were raised for 59 days in a state-of-the-art environmental and temperature-controlled house equipped with temperature, photoperiod, and ventilation automatic regulators with access to feed and water ad libitum. From day 0 to 35 of age, birds were fed with a crumble starter diet, followed by a pellet finisher diet from day 35 to 59. On days 20 and 21, all treatment groups were challenged with *S. agnetis* 908 at 10^4^ colony forming units per mL (CFU/mL) in the drinking water as previously described [9,10,25]. The water supplies were shifted back to tap water after bacterial challenge. On days 11, 33, and 56, blood samples were collected from 6–10 birds per treatment group (Figure 1A). Blood samples were collected in tubes containing EDTA and clot activator solutions, 100 µL of the EDTA-blood was aliquoted in 900 µL of Butterfield’s phosphate diluent (BPD) for microbiological analysis. EDTA-blood samples were used for flow cytometry analysis, and clot activator–blood samples were used to separate blood serum for ELISA analysis. The microbiological analysis was performed using CHROMagar Staph aureus (CHROMagar™, Paris, France) agar plates to enumerate *Staphylococci*.

From day 22, all broilers that died or developed clinical lameness were recorded by date and pen, and birds incapable of standing or walking were diagnosed as lame [1] and then necropsied to assess their BCO lesion severity [1]. The BCO lesions in the proximal femur and tibia were classified: N = femur head and proximal tibia appear entirely normal; FHS = femoral head separation (epiphyseolysis); FHT = femoral head transitional degeneration; FHN = femoral head necrosis (BCO); THN = tibial head necrosis; THNC = tibial head necrosis caseous; THNS = tibial head necrosis severe. Other symptoms include TD = tibial dyschondroplasia [1,6,22,37]. At the end of the experiment on day 59, 10 birds of each group were randomly selected to be necropsied for gross evaluation of tibial and femoral lesions.

### 2.7. Flow Cytometry

All antibodies were purchased from Southern Biotech (Birmingham, AL, USA) unless otherwise specified. Samples were prepared as described previously in Selinger et al., 2012 [38]. Briefly, 20 µL of EDTA-blood was diluted in 980 µL of staining buffer (PBS pH 7.2 + 1% BSA + 0.1% NaN3). Fifty microliters of the diluted blood sample was mixed with fifty microliters of the respective antibody mixture (1:100) and incubated for 45 min at 4 °C protected from light. Three combinations of mouse anti-chicken antibody were used to identify (1) thrombocytes and leukocytes: APC-CD45 (LT40) and FITC-CD41/61 (11C3, Bio-Rad, Hercules, USA), (2) T cells: APC-CD45 (LT40), PE-γδTCR (TCR1), FITC-CD4 (CT-4), and SPRD-D8α (CT-8), (3) B cells, monocytes, and heterophils: APC-CD45 (LT40), FITC-KUL01 (KUL01), and PE-Bu1 (AV20) (Figure 2A). A staining buffer (150 µL) was added, and samples were kept on ice and protected from light until measurement. Samples were analyzed using a BD C6 Accuri flow cytometry machine (Becton Dickinson, San Jose, CA, USA), and results were analyzed using FlowJo v10.8.1 software.

### 2.8. ELISA

The antibody titers against *S. aureus* and *S. agnetis* in blood serum were measured using direct enzyme-linked immunosorbent assay (ELISA). Briefly, wells of 96-well polystyrene microtiter flat bottom plates were coated with 50 µL of the equally mixed *S. aureus* and *S. agnetis* diluted in coating buffer (100 mM NaHCO_3_ buffer, pH 9.6) and incubated at 4 °C overnight. Plates were washed with 200 µL of 0.05% (*v*/*v*) Tween 20-PBS (PBS-T) three times. Next, plates were blocked with 100 µL 5% skim milk solution diluted in PBS-T (blocking solution) at room temperature for 90 min. Sera from birds were diluted 1:500 with blocking solution, and 100 µL of each was added in triplicate to a 96-well plate and incubated for 1 h at 37 °C, while blocking solution was added to negative control (NC) wells. Then, wells were washed three times with 200 µL of PBS-T for 5 min at room temperature. After washing, 100 µL of commercial antibodies of HRP-conjugated anti-chicken IgY (Proteintech^®^, Rosemont, IL, USA), anti-chicken IgM (Invitrogen™, Waltham, MA, USA), and anti-chicken IgA (Invitrogen™, Waltham, MA, USA) were diluted 1:10,000 in blocking solution and added to each well and incubated for 1 h at 37 °C, then washed three times with 200 µL of PBS-T for 5 min at room temperature. Then, 100 µL of 3,3′,5,5′-tetramethylbenzidine (TMB, ThermoFisher) was added and incubated for 15 min at room temperature, followed by the addition of 100 µL of stop solution (2M H_2_SO_4_, Thermo Fisher Scientific, Waltham, MA, SUA). The optical density (OD) at 450 nm of each well was measured using a Cytation 5 multimode microplate reader (Agilent, Santa Clara, CA, USA). The OD values were recorded as the mean of triplicate wells. The ELISA value was standardized as follows: ELISA value = (SampleOD_450_/NC OD_450_), where SampleOD450 and NC OD_450_ represent the OD at 450 nm. All samples were run in triplicates, and the experiments were run two times.

### 2.9. Statistical Analyses

Statistical analyses were conducted using GraphPad Prism version 9.1 (GraphPad Software, San Diego, CA, USA). For comparisons of lameness incidences, Kaplan–Meier survival analysis was used followed by the Mantel–Cox post hoc test. Flow cytometry and ELISA data were analyzed by the Shapiro–Wilk test to verify their normal distribution. Differences in the whole-blood cell populations between the three groups were analyzed by two-way ANOVA followed by Tukey’s post hoc test. To analyze the differences in the antibody response between the three groups, one-way ANOVA followed by Tukey’s post hoc test was used. To compare the probability of lameness or occurrence of femur and tibia lesions between the three groups, a logistic regression was fit in R version 4.2.2, R Foundation for Statistical Computing, Vienna, Austria. Odds ratios between each vaccination treatment and the control were calculated for the odds of lameness or lesions, and the *p*-values of the contrasts were adjusted with the Dunnett adjustment. *p*-values ≤ 0.05 were considered statistically significant.

## 3. Results

### 3.1. eBeam-Treated Whole-Cell Staphylococcus Vaccine Prevented the Development of Lameness in Broiler Chickens

Broiler chicken eggs were vaccinated with vehicle (Control group), formalin-treated *Staphylococcus* (Formalin group), or eBeam-treated *Staphylococcus* (eBeam group), then at days 20 and 21 chickens were challenged via drinking water with *S. agnetis* 908. After the challenge, birds were evaluated twice daily to monitor lameness development (Figure 1A). Our study data showed that compared to the Control group, the eBeam vaccine group presented approximately 50% less lameness, while the Formalin group presented 3% (Figure 1B). Additionally, the eBeam vaccine group presented a significantly lower probability of developing lameness than the Control and Formalin groups (Figure 1C). The odds ratio analysis showed that the probability of developing lameness was similar for Formalin and Control groups (*p*-value > 0.05), but the eBeam group had a significantly lower probability compared to both groups (*p*-value = 0.0008) (Figure 1D). The contrast analysis of the odds ratio of eBeam versus Control groups revealed that the odds of lameness in the eBeam treatment was only 18.5% of the odds of lameness of the Control group, while the Formalin group presented 88.2% of the odds of lameness of the Control group, which was not significantly different from the Control group (Figure 1E).

### 3.2. Staphylococcus eBeam Vaccine Prevents BCO Lesions in Broiler Chicken Femur and Tibia

After challenging the birds with *S. agnetis* 908 on days 20 and 21, all broilers were evaluated twice daily for clinical lameness. All birds that were found dead or developed clinical lameness were necropsied to characterize the proximal femur lesions (Table 1). For the lame birds, all groups presented similar incidences of femoral lesions. Additionally, the data presented no discernible pattern for the BCO lesion severity regarding right or left femoral and tibial lesions.

At the end of the experiment on day 59, 10 birds of each group were randomly selected and necropsied for gross evaluation of tibial and femoral lesions, as well as identification of their lameness severity. The lameness incidence of the eBeam group was significantly lower than that of the Control group (Table 2). The contrast analysis of the odds ratio of the Formalin and eBeam groups compared to control revealed that the eBeam-vaccinated group’s odds of lameness was 4.74% of the odds of the control group (*p*-value = 0.0301), and the Formalin group presented similar probability to develop lameness compared to the Control group (*p*-value > 0.05). However, all three groups presented similar probabilities for the incidences of femoral and tibial lesions. Our data showed that the eBeam vaccine protected broiler chickens from developing lameness, but the eBeam-vaccinated birds presented similar femur and tibia lesion grades as the formalin-vaccinated and control broiler chickens.

### 3.3. eBeam Vaccine Decreases the Time to Develop a More Efficient Immune Response Against Staphylococcus Infection in Broiler Chickens

Blood samples were collected on days 11 (n = 6), 33 (n = 6), and 56 (n = 10) from all three groups for flow cytometry and ELISA analysis. Microbiological analysis of blood was performed at days 33 and 56, and numbers of *Staphylococcus* colonies of blood samples were similar in all three groups. Cytological analysis of whole blood by flow cytometry identified leukocytes as CD45^+^CD41/61^−^ cells. Further analysis characterized the leukocyte populations as: CD4^+^T cells (CD45^+^CD4^+^CD8^−^TCRγδ^−^), CD8^+^T cells (CD45^+^CD4^−^CD8^+^TCRγδ^−^), γδT cells (CD45^+^CD4^−^CD8^+or−^TCRγδ^+^), B cells (CD45^+^Bu1^+^KUL01^−^), monocytes (CD45^+^Bu1^−^KUL01^+^), and heterophils (CD45^+^Bu1^−^KUL01^−^SCC^hi^) (Figure 2A). The results of flow cytometry analysis showed that all groups presented similar absolute cell count concentrations of total leukocytes and all leukocyte populations in the blood at the timepoints of sample collection (Figure 2B).

For antibody titer quantification, ELISA analyses were performed using blood serum. On day 11, all groups presented similar titers of IgM, IgY, and IgA (Figure 3A). On day 33, the eBeam group had significantly higher IgA titer than the Control group and the Formalin group had significantly higher IgM and IgY titers than the Control group (Figure 3B). On the last day of sample collection, Formalin and eBeam groups presented IgM, IgY, and IgA similar to the Control group (Figure 3C). However, the Formalin group showed significantly higher IgY titer at days 33 and 56 than the eBeam group (Figure 3B,C). These results confirmed that both vaccines, formalin- and eBeam-treated *Staphylococcus* cells, were successful in producing a faster humoral immune response compared to the Control group after the chicken was challenged. The eBeam vaccine group presented a significantly higher IgA titer than the Control group in the bloodstream, indicating a higher secretion of IgA in the mucosa. The immunoglobulins present in the mucous, IgA, help to neutralize pathogens before they can infiltrate the mucosa and gain access to the bloodstream. In contrast, IgY is present in the blood serum and can only neutralize pathogens after infiltrating the tissues. These data provide evidence that eBeam vaccine generated a more specific and efficient immune response against *Staphylococcus* than the formalin vaccine.

## 4. Discussion

*Staphylococcus*, more specifically *S. aureus*, presents a broad range of virulence factors, including toxins, capsule, adhesin, and anti-immune system factors, which have been used to create antigen-specific vaccines [39]. However, because of the adaptability of the staphylococcal genome, a large number of strains would need to be tested to ensure that the targeted-antigen vaccine will be broadly protective [17,18]. In contrast, we developed an eBeam-killed *Staphylococcus* vaccine that generated an efficient immune response and prevented the development of clinical lameness in broiler chickens (Figure 1). Our results showed the importance of using a whole-cell vaccine that maintains the integrity of the membrane surface proteins in creating an efficient immune response.

BCO pathognomonic lesions have been observed in clinically non-lame birds [11,40]. Our data showed a significant reduction of lameness incidence in the eBeam-vaccinated group compared to the other two groups; however, the femur and tibia lesion severity for birds that became lame was statistically similar between all treatments. At day 59, necropsy evaluation of surviving birds determined there were a higher number of normal femurs in the Formalin and eBeam groups compared to the Control group (Table 2), suggesting that the eBeam vaccine was able not only to prevent the development of clinical lameness but it may also efficiently prevent BCO lesions in the femur.

The complex tissue response to activate the immune system and effectively contain or eliminate the microbial infection is composed of vascular changes, the release of molecular signaling cascades, and local cellular activities. The cytokines and chemokines released by the local tissue cells will recruit leukocytes and immune complex proteins, like antibodies and complement system proteins from the blood [41,42]. The necropsy data showed that the eBeam vaccine prevented the development of lameness in the broiler chickens (Figure 1), but we did not observe a change in leukocyte populations in blood when comparing the three groups (Figure 2). These data combined suggest that the timepoints for collecting blood samples missed the leukocyte migration generated by the immune response caused by the *Staphylococcus* infection. For future studies, it will be important to consider the collection of the affected tissue and secondary lymphoid tissues to evaluate if the eBeam group presented a different leukocyte response to the *Staphylococcus* challenge compared to the Control group.

Additionally, *Staphylococcus* is known to secrete proteins to block cellular receptors and serum proteins involved in innate immunity, including many complement system inhibitors [39], which may explain our difficulty in detecting the variation of leukocyte populations in the blood. Furthermore, several proteins secreted by S. aureus, such as *Staphylococcus* protein A (Spa), staphylococcal binder of immunoglobulin (Sbi), and staphylococcal superantigen-like protein 10 (SSL10), bind to the human IgG that is deposited on the bacterial cell surface and disrupts IgG-mediated phagocytosis [43,44,45,46]. Staphylococcal superantigen-like 7 (SSL7) was described to interact with IgA and complement C5, inhibiting IgA-FcαRI phagocytosis in human blood [47]. However, a previous study showed that secretory IgA presents higher opsonization and is capable of triggering the H_2_O_2_ release of polymorphonucleated (PMN) leukocytes in the immune response against *Staphylococcus* [48]. Consistent with these studies, our results showed that the eBeam-killed *Staphylococcus* vaccine stimulated the development of an IgA-mediated immune response, which was more efficient in preventing lameness in broiler chickens compared to the formalin-killed vaccine that developed an IgY-mediated immune response (Figure 3).

## 5. Conclusions

Reducing BCO lameness in broiler chicken production is essential to improving chicken welfare and decreasing economic losses in the industry. Our data support the use of *Staphylococcus* treated by eBeam technology to generate an efficacious vaccine to prevent BCO lameness in broiler chickens. We were able to observe a significantly lower incidence of lameness in the eBeam group compared to the other groups. Also, we detected a significantly higher concentration of IgA in the serum of eBeam group chickens compared to the individuals of other groups, supporting that the eBeam-treated *Staphylococcus* vaccine generated a more efficient and effective immune response. Further studies are necessary to increase our knowledge about the cytological and molecular mechanisms of the immune response generated by this vaccine.

## 6. Patents

“A highly effective electron beam (eBeam)-killed multi-bacterial vaccine for mitigating broiler chicken lameness, elevating avian health, improving animal welfare and reducing financial loss”. Provisional Patent Application Filed.

## Figures and Tables

**Figure 1 vaccines-12-01203-f001:**
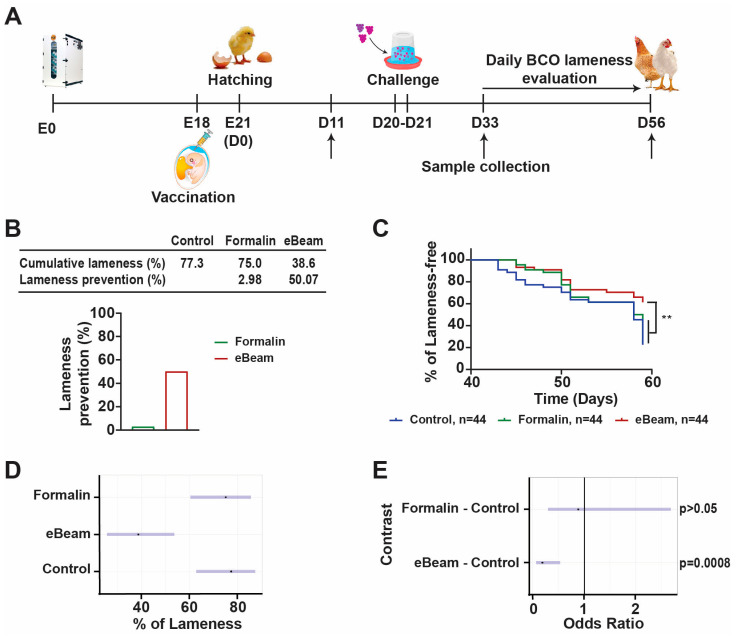
**eBeam vaccine prevents lameness in broiler chickens.** (**A**) Lameness trial experimental strategy. (**B**) Table shows percentages of cumulative lameness and lameness prevention of Formalin and eBeam groups compared to Control group and representative graph of lameness prevention. (**C**) Kaplan–Meier lameness-free curve for broiler chickens. (**D**) Population plot estimates of probability of lameness with 95% confidence interval (blue line). (**E**) Plot of the odds ratios for treatment versus control with 95% confidence interval (blue line) and OR = 1 (black line—indicates no difference). n represents the number of chickens evaluated. ** *p*-value < 0.01.

**Figure 2 vaccines-12-01203-f002:**
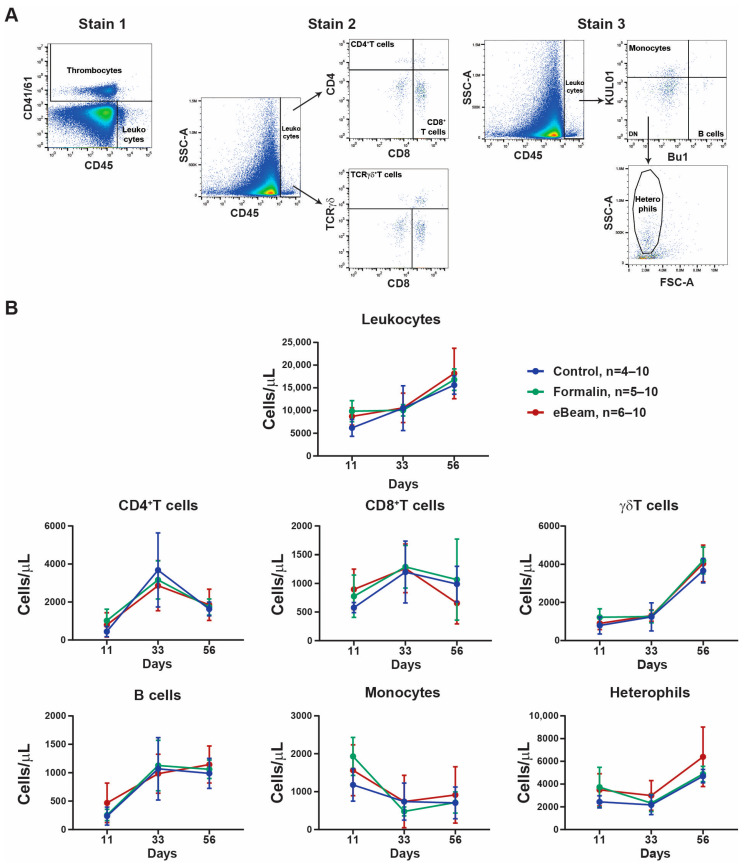
**Leukocyte populations in broiler chickens were similar between the three experimental groups.** (**A**) Representative flow cytometry gating strategy for stains 1, 2, and 3. (**B**) Quantification of absolute cell number of leukocytes (CD45^+^CD41/61^−^ cells), CD4^+^T cells (CD45^+^CD4^+^CD8^−^TCRγδ^−^), CD8^+^T cells (CD45^+^CD4^−^CD8^+^TCRγδ^−^), γδ T cells (CD45^+^CD4^−^CD8^+or−^TCRγδ^+^), B cells (CD45^+^Bu1^+^KUL01^−^), monocytes (CD45^+^Bu1^−^KUL01^+^), and heterophils (CD45^+^Bu1^−^KUL01^−^SCC^hi^) at day 11 (control n = 4; formalin n = 5; eBeam n = 6), day 33 (n = 6), and day 56 (n = 10). n represents the number of chickens evaluated; graphs show means ± SD.

**Figure 3 vaccines-12-01203-f003:**
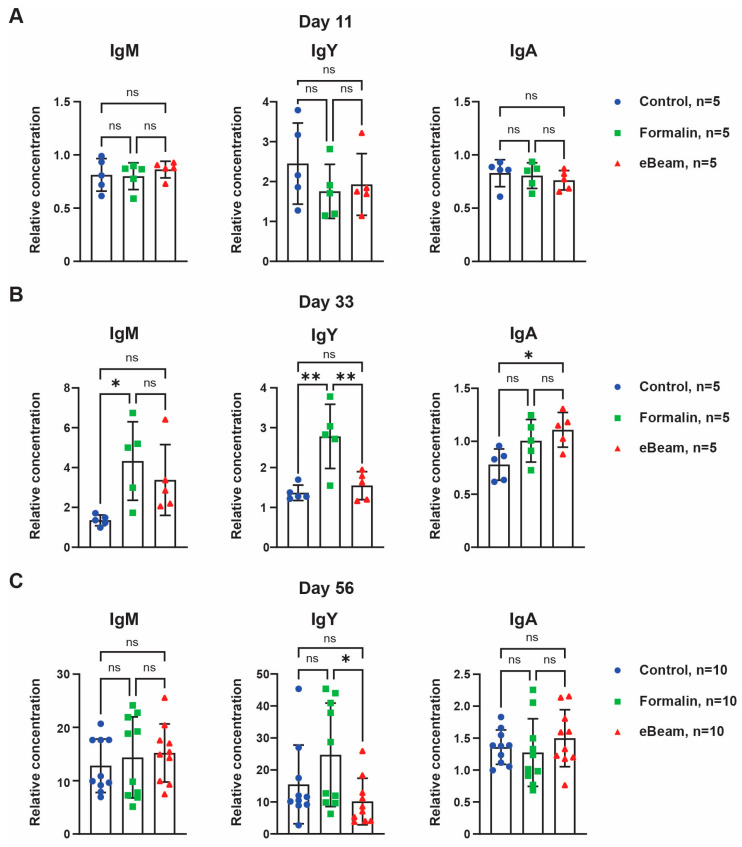
**eBeam vaccine developed a more specific immune response compared to other treatments.** Quantification of the relative concentration of IgM, IgY, and IgA antibodies in the serum of broiler chickens of the three treatments at days 11 (**A**), 33 (**B**), and 56 (**C**). n represents the number of chickens evaluated; graphs show means ± SD; ns *p*-value > 0.05, * *p*-value < 0.05, ** *p*-value < 0.01.

**Table 1 vaccines-12-01203-t001:** Percentage of proximal femoral necropsy incidence during 59 days of experiment for broilers that developed (BCO) lameness within the groups treated with sham (control), formalin vaccine, or eBeam vaccine.

	Right Femur				Left Femur			
Groups	Normal Femur	FHS	FHT	FHS + FHT	Normal Femur	FHS	FHT	FHS + FHT
Control, n = 34	8.82	50.00	41.18	91.18	5.88	61.76	32.35	94.12
Formalin vaccine, n = 33	15.15	30.30	54.55	84.85	15.15	33.33	51.52	84.85
eBeam vaccine, n = 17	5.88	41.18	52.94	94.12	11.76	58.82	29.41	88.24

FHS = femoral head separation (epiphyseolysis); FHT = femoral head transitional degeneration; (BCO).

**Table 2 vaccines-12-01203-t002:** Evaluation of proximal femoral and tibial lesion incidence of broilers at the end of the experiment (day 59) within the groups that were treated with sham, formalin vaccine, or eBeam vaccine.

Groups	Lameness	Normal Femur	FHS	FHT
Control, n = 10	7 ^a^	0	6	6
Formalin vaccine, n = 10	7 ^a^	4	3	8
eBeam vaccine, n = 10	1 ^b^	5	3	5

FHS = femoral head separation (epiphyseolysis); FHT = femoral head transitional degeneration. ^a,b^ indicates statistical significance (same letter *p*-value >0.05; different letters *p*-value < 0.05).

## Data Availability

Raw data supporting the conclusions of this manuscript will be available upon request.

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
