# Peer review of "Electron-Beam-Killed Staphylococcus Vaccine Reduced Lameness in Broiler Chickens"

_vaccines, 2024, doi:10.3390/vaccines12111203_

Round 1
Reviewer 1 Report
Comments and Suggestions for Authors
The authors present an interesting study, that reports the use of electron beam (eBeam) inactivate bacterial cells for the purposes of immunising chickens against the development of bacterial chondronecrosis with osteomyelitis (BCO) associated with Staphylococcus aureus and S. agnetis infection. The study reports that the bacteria inactivated in this manner were able to elicit protective immune responses in challenge studies. The authors present evidence of the eBeam inactivated bacteria eliciting specific IgA responses and protected against BCO. The results suggest that inactivation the bacteria with eBeam protects the antigenicity of the protective epitopes of the pathogens of interest.
The methods of the study are well described and would enable the replication of the study.
The results are well presented and support the discussion and conclusions.
Can the authors confirm that no adjuvants were used in any of their inactivated bacterial formulations?
Line 26 suggest revision “is associated with”
Line 28 suggest revision “immune deficiency resulting from stress, or inflammation”
Line 39 Please review the keywords. Suggest no abbreviations.
Line 47 consider revision “and increasing feed conversion ratios”
Line 65-66 Suggest adding a supporting citation(s) for infections in children and other at risk groups.
Line 69 suggest revision “in stimulating an adequate”
Line 82 Suggest adding a supporting citation(s)
Line 99 suggest details of the source of these isolates. For example, I assume they were isolated from broiler chickens affected by BCO?
Lines 116 to 119 What temperature were the bacterial cells transported at? Suggest specifying this in the text.
Line 255 Figure 1E – Was the likelihood of BCO in the eBeam vaccinated group significantly lower compared to the sham vaccinated group? It would appear they maybe as the 95% CI do not span 1. If this protective effect was significant the graph should indicate this.
Line 333 Figure 3 The legend and the y-axis refer to “relative expression” – I do not think this the correct term. Based on the description in the methods, “relative absorbance”, might be more appropriate.
Line 355-357 – Please review this sentence and revised as necessary, it does not seem to make sense to me.
Line 361-369 – Typically, in immunisation studies we do not quantify the immune response by changes in the lymphocyte populations. Rather isolation of these cells and simulating with antigen is used to assess the development of specific immune responses. Thus, the response of specific populations of lymphocytes to antigen stimulation would be a better way of quantifying the eliciting of cellular responses to vaccination. Please review this text.
Line 416 On line 396 the authors suggest that a patent has been filed regarding the content of this manuscript. If any of the authors named on this manuscript are also named inventors of this provisional patent, I would suggest this at the very least a perceived conflict of interest, if not an actual conflict of interest.
Comments on the Quality of English LanguageSee comments to authors.
Reviewer 2 Report
Comments and Suggestions for Authors
This is an interesting experimental study focusing on the development of an effective vaccine against Staphylococcus infections in broiler chickens. Staphylococcus infections are indeed a problem in poultry farming and given the high economic burden of such infections -mentioned by the authors (economic loss of 150$ billion each year)- the paper merits publishing. The authors provide an effective way to produce such a vaccine using eBeam technology as a method to inactive staphylococcus species without denaturing bacterial properties which could cause disruption of lineear or conformational epitopes which are necessary for the development of a vaccine which induces humoral immunity (mainly through IgA) which is necessary to protect the host from colonization and infection. It would be interesting though if the authors could provide any data by challenging broilers with different Staphylococcus strains so as to prove the existence of cross immunization or whether the vaccine has any effect in the colonization of broilers by infective strains. Otherwise the paper is well written and I have no further comments.
Comments on the Quality of English LanguageThe English language is fine.
